

# Biomarker selection depends on gene function and organ: the case of the cytochrome P450 family genes in freshwater fish exposed to chronic pollution

Jorge Cortés-Miranda[1], Noemí Rojas-Hernández[1], Gigliola Muñoz[2], Sylvia Copaja[2], Claudio Quezada-Romegialli[3], David Veliz[1,4] and Caren Vega-Retter[1]

[1] Departamento de Ciencias Ecológicas, Facultad de Ciencias, Universidad de Chile, Santiago, Región Metropolitana, Chile
[2] Departamento de Química, Facultad de Ciencias, Universidad de Chile, Santiago, Region Metropolitana, Chile
[3] Laboratorio de Genómica y ADN ambiental, Facultad de Ciencias Agronómicas, Universidad de Tarapacá, Arica, Arica y Parinacota, Chile
[4] Centro de Ecología y Manejo Sustentable de Islas Oceánicas., Coquimbo, Coquimbo, Chile

Corresponding author
Caren Vega-Retter,
carenvega@uchile.cl

## ABSTRACT

Pollution and its effects have been of major concern in recent decades. Many strategies and markers have been developed to assess their effects on biota. Cytochrome P450 (CYP) genes have received significant attention in this context because of their relationship with detoxification and activation of exogenous compounds. While their expression has been identified as a pollution exposure biomarker, in most cases, it has been tested only after acute exposures and for CYP genes associated with exogenous compounds. To elucidate CYP gene expression patterns under chronic pollution exposure, we have used the silverside *Basilichthys microlepidotus* as a model, which inhabits the Maipo River Basin, a freshwater system with different pollution levels. We performed next-generation RNA sequencing of liver and gill tissues from polluted and non-polluted populations. We found most CYP genes were not dysregulated by pollution, and the seven genes that were present and differentially expressed in liver and gill were mainly downregulated. Three CYP genes associated with exogenous compounds showed differential expression in the gill, while four CYP genes associated with endogenous compounds showed differential expression in the liver. The findings presented here highlight the importance of CYP genes, his family, tissues and his interaction in the context of pollution biomarkers use.

## INTRODUCTION

In the context of global change, pollution is one of the more relevant factors affecting biota (*Sage, 2020*). Pollution is widely distributed in ecosystems worldwide, with freshwater

ecosystems being one of the most affected by human activities (*Jackson et al., 2016*). Researchers have invested significant effort in quantifying pollution levels in freshwater ecosystems, from water quality monitoring to biological assessments, to understand their ecological impacts on flora and fauna inhabiting these systems (*Zorita et al., 2007*).

Recent biological analyses have used different approaches to quantify the effects of pollutants on biota. These approaches range from individual to ecosystem levels. At the individual and population levels, biochemical and genetic responses have been extensively studied to characterize molecular-level strategies to cope with pollution from an evolutionary perspective (*Fisher & Oleksiak, 2007*; *Bélanger-Deschênes et al., 2013*; *Ben-Khedher et al., 2013*). These approaches include exploratory omics methods, such as transcriptomics, genomics, and metabolomics, to identify pollution-related genotypes and biochemical responses. In this context, transcriptomics has been applied in multiple species to identify differential expression of pollution-associated genes (*Baillon et al., 2015*; *Vega-Retter et al., 2018*). Differentially expressed genes have been reported to be dysregulated between organs of organisms exposed to pollution (*Rhee et al., 2009*; *Ragusa et al., 2017*).

*Goldstone et al. (2006)* developed the concept of the chemical defensome. It was defined as a group of genes from different families that act as a network to cope with chemical stress and maintain chemical homeostasis in organisms. Among them, genes related to oxidative biotransformation, such as aldehyde dehydrogenases, flavoprotein monooxygenases, and cytochrome P450s (CYPs), have been described (*Goldstone et al., 2006*; *Eide et al., 2021*). The CYP superfamily comprises genes encoding enzymes that detoxify or activate organic pollutants or that act on endogenous molecules (*Nebert et al., 2004*; *Uno, Ishizuka & Itakura, 2012*). These genes have been found in diverse organisms, including fish species (*Eide et al., 2021*), and have been classified into families catalyzing exogenous or endogenous molecules for fish (*Uno, Ishizuka & Itakura, 2012*). Those acting on exogenous compounds have received particular attention in pollution research. Indeed, the CYP family 1 subfamily A (CYP1A) gene and its encoded enzyme have been proposed as a pollution biomarker (*Goksøyr, 1995*; *Lee & Yang, 2008*). Therefore, CYP1A's expression patterns have been extensively explored. *Wong et al. (2001)* reported CYP1A1 expression levels in different tissues of tilapia (*Oreochromis mossambicus*) exposed to coastal sediments, showing increased expression levels in the liver and intestines compared to other tissues. *Yuan et al. (2013)* reported that basal CYP gene expression levels differed among tissues in the rare Chinese minnow fish *Gobiocypris rarus* after benzo[a]pyrene (BAP) exposure, with CYP1A, CYP family 1 subfamily B member 1 (CYP1B1), and CYP family 1 subfamily C member 1 (CYP1C1) showing strong upregulation in the liver, gills, and intestine. However, to our knowledge, the expression patterns of CYP family categories (endogenous and exogenous) in different tissues have not been explored. However, they could be important for understanding an organism's adaptation to pollution in a biomonitoring program.

One example of freshwater pollution is the Mediterranean-type Maipo River Basin in Central Chile. This catchment has been mainly affected by pollution related to domestic and agricultural activities (*Dirección General de aguas, 2004*) associated with the large

population inhabiting this basin, which represents almost 40% of the Chilean population (*Instituto Nacional de Estadística (INE), 2017*). Different fishes inhabit this catchment, including the endemic silverside *Basilichthys microlepidotus*, which currently has a vulnerable conservation state (*Ministerio del Medio Ambiente, 2022*). This species inhabits lakes and rivers from 28 °S to 39 °S (*Veliz et al., 2012*) and is macrophagic, feeding on small invertebrates, insect larvae, detritus, and filamentous algae (*Bahamondes, Soto & Vila, 1979*; *Duarte et al., 1971*). Its reproductive period is from August to January (*Comte & Vila, 1992*), and it shows different populations within and between river basins in central Chile (*Quezada-Romegialli, Fuentes & Veliz, 2010*), recently colonizing new areas in the Maipo River Basin (*Cortés-Miranda et al., 2022*).

*Vega-Retter et al. (2014)* reported five genetically different populations inhabiting areas with different contamination levels, identifying three populations inhabiting sites categorized as non-polluted (Isla de Maipo–Peñaflor, San Francisco de Mostazal–Maipo, and Puangue) and two populations inhabiting sites categorized as polluted (Melipilla and Pelvin). Individuals from polluted sites showed evidence of pollution-related selection in an amplified fragment length polymorphism analysis (*Vega-Retter, Vila & Véliz, 2015*). Following transcriptome characterization (*Vega-Retter & Véliz, 2014*), genes related to apoptotic processes and carcinogenesis were found to be differentially expressed in the livers of individuals inhabiting polluted sites compared to non-polluted sites (*Vega-Retter et al., 2018*; *Veliz et al., 2020*). Overall, these findings suggest that *B. microlepidotus* is a suitable species for studying the effects of pollution in the Maipo River Basin.

In this study, we performed next-generation RNA sequencing (RNA-seq) on the gills and liver of *B. microlepidotus* individuals inhabiting polluted and non-polluted sites to gain insight into gene expression patterns in these natural populations exposed to chronic pollution. These organs were chosen for their relationship with pollution; the gills are directly exposed to pollutants, and the liver is related to all the physiological and biochemical pathways that pollutants could alter (*Zeitoun & Mehana, 2014*). Due to their use as biomarkers of pollution exposure, we focused our analysis on the expression of the CYP genes with endogenous and exogenous compound targets in both tissues.

# MATERIALS AND METHODS

## Sampling sites and sample collection

This research was conducted in the spring of 2016 and sampled four sites in the Maipo River Basin, Chile. Two sites were characterized in previous studies (*Vega-Retter et al., 2014*; *Vega-Retter et al., 2018*) as historically non-polluted (San Francisco de Mostazal (SFM) (33°58′19.97″S, 70°42′56.49″W) and Isla de Maipo (IM) (33°44′58″S, 70°53′26″W)) and two as polluted (Melipilla (MEL) (33°42′49,988″S, 71°12′39,13″W) and Pelvin (PEL) (33°36′21″S, 70°54′33″W); Fig. 1). In the previous studies, the non-polluted sites have shown good water quality related to a low-density population and low industrial development nearby. In recent years, increasing urbanization and industrial development close to the IM site has decreased the water quality. In contrast, the polluted sites are downstream of wastewater plants and historical industrial water discharges (*Gomez, De La Maza & Melo, 2014*).

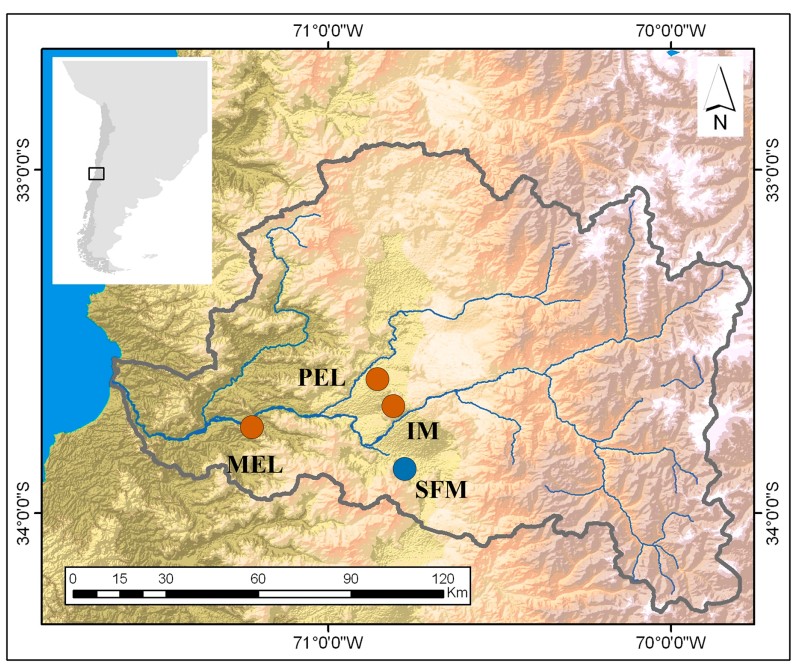

**Figure 1 Map of the study sites in the Maipo River Basin.** Melipilla (MEL), Pelvin (PEL), Isla de Maipo (IM) (orange) and San Francisco de Mostazal (SFM) (blue). Modified from *Veliz et al. (2020)*.

Twenty-four *B. microlepidotus* individuals were collected by electrofishing, six per sampling site. Sampling was performed post-reproductive period; thus, individual sex cannot be determined using gonads. The mean fish weight was 5.04 ± 0.90 g for the IM site, 1.90 ± 0.61 g for the MEL site, 5.41 ± 2.52 g for the PEL site, and 16.03 ± 7.97 g for the SFM site. The mean total length was 10.03 ± 0.50 cm for the IM site, 7.16 ± 0.69 cm for the MEL site, 10.27 ± 1.57 cm for the PEL site, and 14.27 ± 2.16 cm for the SFM site. The fish were sacrificed immediately by neck-breaking, and the liver and gills were removed in the field and stored in liquid nitrogen for subsequent RNA extraction.

All the protocols used in this study were approved by the Ethics Committee of the Universidad de Chile and complied with existing laws in Chile (Resolución Exenta No. 3078 Subsecretaria de Pesca).

## Physical and chemical characterization of the sampling sites

The surface water and sediment were physically and chemically characterized to determine the current pollution level at each sampling site, focusing on micronutrients, macronutrients, and metals as an approximation to the complex mixture of contaminants in the river. Three samples of surface water (1 L) and sediment (1 kg) were taken at each site. The water samples were collected in the water column using Nalgene vials and stored at 4 °C for at most 48 h before analysis. In addition, physical and chemical parameters such as electrical conductivity (EC), temperature, dissolved solids (DS), and pH were measured three times *in situ* using a multiparameter device (Hanna Instruments, Woonsocket, RI, USA).

## Superficial water characterization

For water samples, nitrite ($NO_2^-$) is determined by the formation of a reddish-purple azo dye by diazotization-coupling reaction of sulfanilamide with N-(1-naphthyl)-ethylenediamine dihydrochloride (NED dihydrochloride) at pH between 2.0 and 2.5 and was quantified by spectrophotometry at 543 nm (UV-1700; Shimadzu; Kyoto, Japan) (*Sadzawka et al., 2006*).

In the case of ammonium ($NH_4^+$), it was quantified by Indophenol blue method, which is a colorimetric method (*Solórzano, 1969*).

Boron (B) was quantified by the technique proposed by Berger and Truog (*Sadzawka et al., 2006*), measuring the absorbance at a length of 420 nm (UV-1700; Shimadzu; Kyoto, Japan).

Phosphate ($PO_4^{3-}$) was quantified by a colorimetric method that is based on the formation of a hetero-polyacid with the vanado-molybdic reagent, and a subsequent measure of absorbance at a length of 880 nm (UV-1700; Shimadzu; Kyoto, Japan).

Carbonate ($CO_3^{2-}$), bicarbonate ($HCO_3^-$), chloride ($Cl^-$), nitrate ($NO_3^-$) and sulfate ($SO_4^{2-}$) were determined by Ion Exchange High Performance Liquid Chromatography (IE-HPLC) Isocratic HPLC Pump with a conductivity detector, using anionic (IC-Pak HJC) columns. Isocratic conditions were used with injection volume 50 μL and mobile phase flow 1.2 mL/min. The mobile phase was composed of concentrated borate/gluconate prepared with 34 g boric acid (Merck, Lebanon, NJ, USA), 23.5 mL gluconic acid (Merck, Lebanon, NJ, USA), 8.6 g lithium hydroxide (Merck p. a., Lebanon, NJ, USA) and 250 mL glycerin (Merck, Lebanon, NJ, USA), diluted to 500 mL with MilliQ deionized water. To 20 mL of this solution, we added 120 mL acetonitrile (Merck HPLC grade, Lebanon, NJ, USA) and diluted to 1 L with MilliQ deionized water. Analytes identification was determined by retention time compared with standards (Merck titrisol, Lebanon, NJ, USA). Anion concentrations were estimated using calibration curves generated with standards. Data quality was monitored by measuring element concentration in procedural blanks and synthetic preparations of deionized water (MilliQ) with analytes (*Copaja, Núñez & Veliz, 2014*).

For, sodium ($Na^+$), potassium ($K^+$), calcium ($Ca^{2+}$), magnesium ($Mg^{2+}$) were quantified by atomic absorption spectroscopy (AA-6880; Shimadzu, Kyoto, Japan) after filtering the samples using a cellulose nitrate filter with a 0.45 μm pore diameter (Sartorius, Göttingen, Germany) according to *Clesceri, Greenberg & Eaton, 2005*. Total solids (TSs) were quantified by taking a 100 mL aliquot of water, evaporating the water until dry, and weighing the resulting solid. DSs were quantified by taking a 100 mL aliquot, filtering it through a membrane with a pore diameter of 0.45 μm, evaporating the water until dry, and weighing the resulting solid. Finally, suspended solids (SSs) were calculated as TS minus DS.

Water samples were fixed with 2% $HNO_3$ (Suprapur Merck, Darmstadt Germany) to quantify iron (Fe), copper (Cu), zinc (Zn), cadmium (Cd), manganese (Mn), nickel (Ni), aluminum (Al), molybdenum (Mo), lead (Pb), mercury (Hg), arsenic (As), and chromium (Cr) concentrations using an atomic absorption spectrophotometer (AA-6880; Shimadzu, Kyoto, Japan), given the reported effects of these metals on biota.

The dissolved oxygen (DO) concentration was estimated by taking three water samples from each sampling site in 200 mL polycarbonate bottles (Nalgene, Rochester, NY, USA), fixing them with $MnSO_4$ and alkaline iodide, and analyzing them using the Winkler method according to *Strickland (1968)*.

The oxygen used by microorganisms to decompose organic waste was measured as the biochemical oxygen demand ($BOD_5$). Briefly, three 300 mL flasks of water from each site were incubated at 20 °C for 5 days before oxygen quantification, as described by *Clesceri, Greenberg & Eaton, 2005*.

### Sediment characterization

At each sampling site, three 1 kg sediment samples were collected from the top 10 cm of the surface using a plastic scoop and polyethylene bag (*Copaja & Muñoz, 2018*). Samples were stored at 4 °C, subsequently dried at room temperature in polyethylene trays, and then sieved into two fractions: a coarse fraction with a particle size of <2 mm for physical and chemical characterization and a fine fraction with a particle size of <0.063 mm for metal quantification.

The physicochemical variables measured in sediment samples were EC, pH, B, $PO_4^{3-}$, N, $Ca^{2+}$, $Mg^{2+}$, and $Na^+$. EC and pH were determined by potentiometric methods (sediment: water = 1:2.5). The *Berger & Truog (1939)* method was used to quantify B with a spectrophotometer (Pharmaspec 1700; Shimadzu, Kyoto, Japan) at 420 nm. The Kjedhal digestion method (*Bremner, 1960*) was used to quantify total N. $Ca^{2+}$, $Mg^{2+}$, and $Na^+$ were quantified by preparing a saturation extract using 50 g of each sample, adding water to saturate the sample in a 1:1 ratio. The extract was left overnight and then centrifuged to collect the supernatant. Quantification used 10 mL of the supernatant using the same procedure as for superficial water. The same metals quantified in the water were analyzed in the sediment: Cu, Fe, Zn, Cd, Mn, Ni, Al, Mo, Pb, Hg, As, and Cr. The total fraction of each metal was obtained by digesting 0.25 g of sediment with 10 mL of nitric acid (Suprapur; Merck, Darmstadt, Germany) in a high-resolution microwave oven (MarsXpress) using the following conditions: power, 800 W; tower, 100%; time, 11 min; temperature, 175 °C; hold, 15 min; cooling, 15 min. This protocol was based on the US Environmental Protection Agency's method 3051,Washington, D.C., US (*Blakemore, Searle & Daly, 1987*). Metal concentrations were determined by flame atomic absorption spectrometry using an atomic absorption spectrometer (AA-6800; Shimadzu, Kyoto, Japan) with an air-acetylene flame.

The sampled sites were categorized and compared with the *Vega-Retter et al. (2014)* categorization using a principal component analysis (PCA) performed with the fviz_pca_biplot function of the *facto-extra* package (*Langmead & Salzberg, 2017*) of the R statistical software (v.4.1.0; *R Core Team, 2023*). The PCA was performed with all physical and chemical data from the four sampling sites (36 environmental variables) to detect relationships among sites. The PCA results showed that the water quality of the IM site had considerably deteriorated in recent years. Therefore, this site was considered polluted for this study. A second PCA was performed using ten physicochemical parameters (EC, pH, DS, DO, $NO_2^-$, $NH_4^+$, $Na^+$, $K^+$, $Ca^{2+}$, and $Mg^{2+}$) also included in historical physicochemical

surface water data for 2007, 2011, and 2016 for the sampling sites. The second PCA aimed to evaluate the physicochemical stability of the sampling sites.

## RNA isolation and RNA-seq

RNA-seq was performed to quantify transcript numbers and determine differential expression between non-polluted (SFM) and polluted (MEL, PEL, and IM) sites. Total RNA was extracted using the PureLink RNA Mini Kit (Ambion; Life Technologies, CA, USA) according to the manufacturer's instructions and sent to Genoma Mayor Sequencing Services (Santiago, Chile), where it was purified to retain only mRNA. RNA quality and quantity were determined using an Agilent 2100 Bioanalyzer (Agilent Technologies, Santa Clara, CA, USA). Samples with an RNA integrity number >7 (*Schroeder et al., 2006*) were subjected to 2 × 100 bp sequencing on an Illumina HiSeq 4000 system (San Diego, CA, USA). Four PEL samples were discarded due to RNA integrity: three liver and one gill sample. Twenty-one liver and twenty-three gill samples were sequenced. The raw sequencing data deposited in SRA database of NCBI with the data accessions BioProject ID: PRJNA1033453 and the BioSample accessions: SAMN38033975–SAMN38034018.

Adapters were removed, and raw reads were filtered using Trim Galore (https://www.bioinformatics.babraham.ac.uk/projects/trim_galore/), prinseq-lite.pl (http://prinseq.sourceforge.net/manual.html), and Cutadapt (*Martin, 2011*). We removed reads with (i) low base quality (qbase ≤ 5), (ii) ≥10% ambiguous bases, (iii) mean Phred score < Q30, and (iv) length < 50 bp.

The *de novo* assembly of gill and liver transcriptomes was performed with the Bridger software (*Chang et al., 2015*) with the following parameters: k-mer length = 25, minimum k-mer seed coverage = 2, and minimum k-mer seed entropy = 1.5. Isoforms were not considered, and contigs with lengths > 200 bp were retained. Clustering was performed with the CD-HIT software (http://weizhongli-lab.org/cd-hit/) to generate a set of non-redundant contigs. An identity cut-off threshold of 80% was used. Non-redundant contigs from both assemblies were annotated using the Blast2GO software (*Conesa et al., 2005*). First, a BLAST search was performed using Blast2GO's blastx function and a subset of vertebrate sequences in the US National Center for Biotechnology Information's non-redundant database with a threshold e-value of $1 \times 10^{-6}$. In addition, the InterPro database was used for annotation. Second, contigs were mapped to identify the gene ontology (GO) terms associated with them, including biological processes, cellular components, and molecular function. Finally, GO terms with an e-value threshold of $<1 \times 10^{-6}$ were retained for annotation.

## Differential expression

Differentially expressed CYP genes were detected by mapping transcripts from each individual against the *de novo* assembly of each tissue using the Bowtie2 software (*Langmead & Salzberg, 2017*) with the following command line flags: -q –phred33 –sensitive –dpad 0 –gbar 99999999 –mp 1,1 –np 1 –score-min L,0, −0.1 −I 1 −× 1000 –no-mixed –no-discordant -p 6 -k 200. BAM files were used to estimate expression levels using the RSEM software (v.1.3.3; *Li & Dewey, 2011*) with the following command line flags: -p 6

–paired-end –calc-pme –calc-ci –ci-memory 30000 –sort-bam-by-coordinate. Finally, differential expression between polluted (MEL, PEL, and IM) and non-polluted (SFM) sites was quantified using the DESeq2 software (*Love, Huber & Anders, 2014*) implemented as a package in the R statistical software (v.4.1.0; *R Core Team, 2023*). Three independent runs were performed for each tissue, comparing all possible pairs of polluted and non-polluted sites (MEL-SFM, PEL-SFM, and IM-SFM) and retaining contigs with a |log fold change (Log$_2$FC)| ≥ 1 and false discovery rate < 0.05. The statistical power of this experimental design was estimated for each organ using the RNASeqPower package (*Therneau, Hart & Kocher, 2023*) in the R statistical software (v.4.1.0; *R Core Team, 2023*).

## CYP gene identification and classification

Differentially expressed CYP genes were identified based on Blast2GO's BLAST results. First, all differentially expressed CYP genes in at least one polluted site and one tissue were considered, and their expression pattern was plotted as Log$_2$FC with the SFM site as the reference. Next, only CYP genes present in *de novo* liver and gill assemblies whose general function was known were retained. The retained CYP genes were classified according to their action on endogenous or exogenous compounds (*Uno, Ishizuka & Itakura, 2012*). To test for differential expression of CYP genes, we used a generalized linear model (GLM) analysis considering differential expression as the dependent variable, sites as the independent variable, and binomial negative as the data distribution. The analysis was run on normalized gene counts using the MASS (v.7.3-60) package (*Venables & Ripley, 2002*) of the R statistical software (v.4.1.0; *R Core Team, 2023*). Additionally, Tukey's test was used to identify significant differences between pairs of sites.

# RESULTS

## Physical and chemical characterization of the sampling sites

Four sampling sites were chosen for this study based on previous studies (*Vega-Retter & Véliz, 2014*; *Vega-Retter, Vila & Véliz, 2015*). Two sites had historically been considered polluted (MEL and PEL), and two non-polluted (SFM and IM; Fig. 1). These sites are inhabited by genetically independent *B. microlepidotus* populations. Thirty-six parameters were used to assess pollution levels at the sampling sites since the concentrations of certain metals in the superficial water (Cu, Zn, Cd, Mn, Ni, Al, Mo, Pb, Hg, and As) and sediment (Cd, Al, Mo, Pb, Hg, and As) were below the detection limit. The parameters used were pH, EC, DO, BOD$_5$, TS, DS, SS, NO$_2^-$, NO$_3^-$, B, PO$_4^{3-}$, Na$^+$, K$^+$, Ca$^{2+}$, Mg$^{2+}$, NH$_4^+$, HCO$_3^-$, CO$_3^{2-}$, Cl$^-$, SO$_4^{2-}$, Fe, and Cr in superficial water, and EC, pH, B, PO$_4^{3-}$, N, Ca$^{2+}$, Mg$^{2+}$, Na$^+$, Cu, Fe, Zn, Mn, Ni, and Cr in sediments. These data were analyzed using a PCA to compare sampling sites.

PCAs of surface water physical and chemical characteristics for 2007, 2011, and 2016 and surface water and sediment physical and chemical characteristics for 2016 at the studied sites are shown in Fig. 2. For the historical physical and chemical data (Fig. 2A), the first two principal components (PCs) explained 62.09% of the total variance. PC1 (41.54% of total variance; eigenvalue = 4.15) segregated the SFM site from the other sampling sites. Three variables had their highest loading (L) in PC1: Ca$^{2+}$ (L = 0.426), K$^+$ (L = 0.408), and
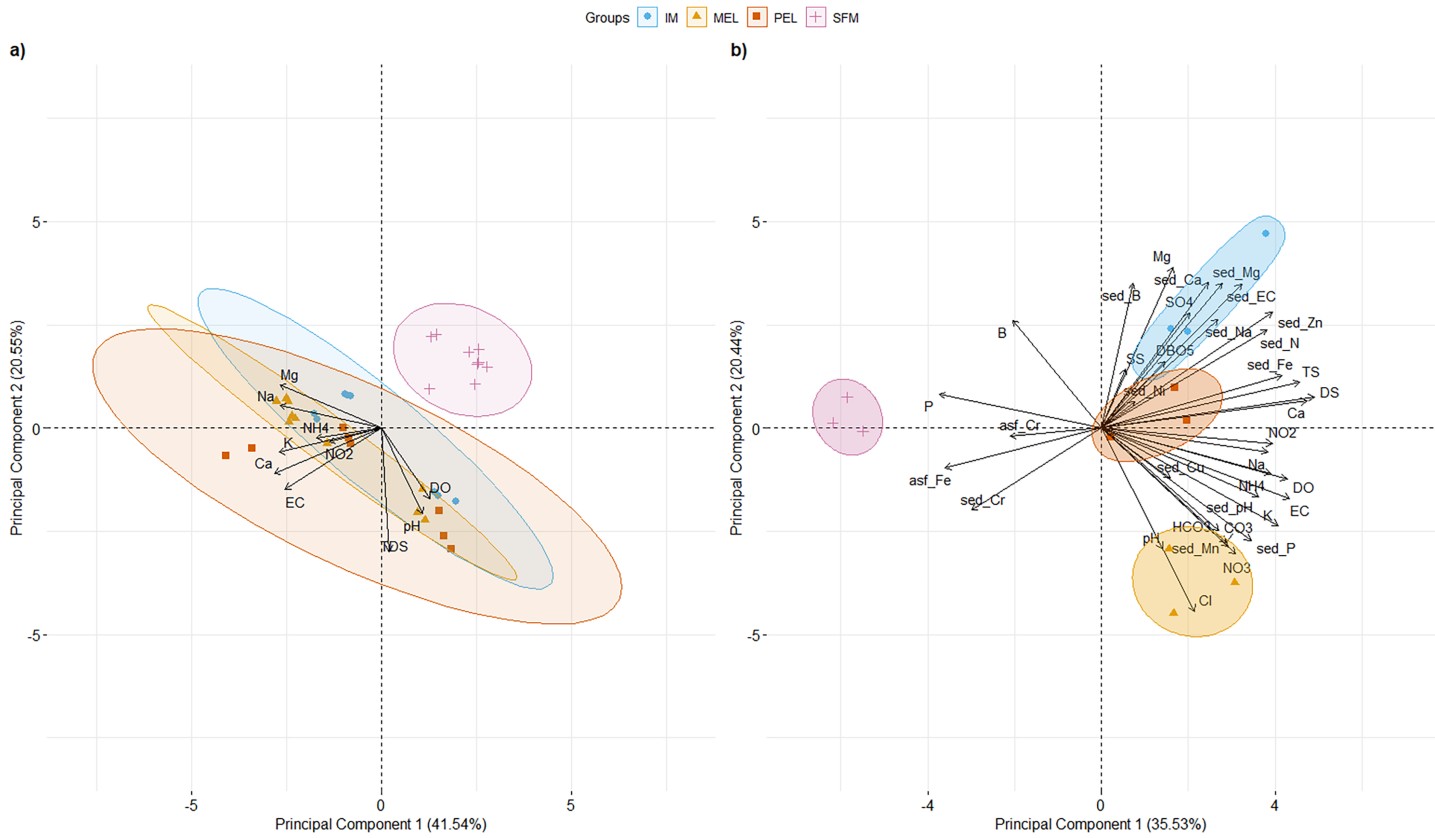

**Figure 2** PCA of the 10 and 36 physical and chemical variables measured in surface water for historical data (A), and water and sediments for 2016 data, including metals (B) for the sampling sites in the Maipo River Basin. Melipilla (MEL), Pelvin (PEL), Isla de Maipo (IM) and San Francisco de Mostazal (SFM). Detail of physical and chemical parameter abbreviations are in Material and Methods. (A) Historical physical and chemical data PCA. (B) physical and chemical data PCA from 2016. Also, metals present in surface water are represented with asf_string and chemical parameters of sediment are represented with sed_string.

Na$^+$ (L = 0.405). Three variables had their highest L in PC2 (20.55% of total variance; eigenvalue = 2.55): DS (L = 0.644), pH (L = 0.448), and DO (L = 0.369).

For 2016 data (Fig.2B), the first two PCs explained 55.96% of the total variance. PC1 (35.53% of total variance; eigenvalue = 12.79) clearly segregated the SFM site from the other sampling sites (IM, MEL, and PEL). Three variables had their highest L in PC1: DS (L = 0.26), Ca$^{2+}$ (L = 0.249), and TS (L = 0.242). Three variables had their highest L in PC2 (20.44% of the total variance; eigenvalue = 7.35): Cl$^-$ (L = 0.29), Mg$^{2+}$ (L = 0.277), and sed_Ca$^{2+}$ (L = 0.251). Values of the main physical and chemical parameters for the historical and 2016 data are shown in Table 1.

Five parameters had values higher than the standards of Decree 53 (*Decreto 53, 2014*; Ministerio del Medio Ambiente (MMA)) at the IM, MEL, and PEL sites in the 2016 data, and two parameters had higher values than the standards at the SFM site. In both the historical and current data, SFM was segregated from the other sampling sites, showing the temporal stability of the physical and chemical patterns found in the 2016 data.

**Table 1 Concentration of the most relevant physical and chemical parameters, related to PCA.**

**2007–2016**

| Site | $Ca^{2+}$ (mg/L) | $K^+$ (mg/L) | $Na^+$ (mg/L) | TDS (mg/L) | DO (mg/L) | EC (µS/cm) |
|------|------|------|------|------|------|------|
| IM | 136 ± 40.8 | 3.64 ± 0.783 | 56.6 ± 33.4 | 851.38 ± 361.09 | 8.20 ± 1.26 | 1,058.63 ± 219.79 |
| MEL | 138 ± 34.4 | 4.65 ± 1.26 | 73.9 ± 40.1 | 775.5 ± 209.16 | 7.82 ± 2.45 | 1,235.6 ± 192.34 |
| PEL | 146 ± 39.2 | 3.31 ± 1.99 | 31.2 ± 27.7 | 919.5 ± 409.39 | 10.87 ± 1.22 | 1,198.63 ± 112.18 |
| SFM | 38.6 ± 14.2 | 1.44 ± 0.866 | 18.8 ± 16 | 286.11 ± 117.89 | 8.58 ± 2.40 | 408.33 ± 83.43 |

**2016**

| Site | DS (mg/L) | $Ca^{2+}$ (mg/L) | TS (mg/L) | $Cl^-$ (mg/L) | $Mg^{2+}$ (mg/L) | sed_$Ca^{2+}$ (mg/L) |
|------|------|------|------|------|------|------|
| IM | 1,099 ± 11.9 | 107 ± 2.18 | 1,282 ± 92.1 | 10.77 ± 0.78 | 1.66 ± 0.001 | 1.29 ± 1.01 |
| MEL | 918 ± 47 | 92.6 ± 7.68 | 1,044 ± 106 | 152.59 ± 43.92 | 0.99 ± 0.02 | 0.12 ± 0.14 |
| PEL | 1,131 ± 46.2 | 118 ± 3.46 | 1,388 ± 245 | 62.77 ± 2.22 | 1.01 ± 0.001 | 0.10 ± 0.01 |
| SFM | 261 ± 90.5 | 36.9 ± 7.55 | 422 ± 111 | 10.77 ± 0.78 | 1.06 ± 0.12 | 0.014 ± 0.004 |

Note:
Mean values were obtained from 8 to 10 replicates per site for historical data (2007–2016) and 3 replicates per site for 2016 data.

**Table 2 Summary statistics of the assembles for both liver and gill of the silverside *B. microlepidotus*.**

| Statistic | Liver | Gill |
|------|------|------|
| *Raw reads* | 344,531,301 | 369,391,488 |
| *Clean reads* | 328,822,670 | 368,882,159 |
| *Assembled contigs* | 74,773 | 125,864 |
| *NR contigs* | 63,424 | 105,131 |
| *Mean length NR contigs* | 1,208.2 | 1,219.39 |
| *Mean %GC NR contigs* | 45.58 | 44.77 |
| *N50 NR contigs* | 2,518 | 2,745 |
| *Largest NR contig* | 27,527 | 29,984 |
| *Mean % Mapped* | 90.68 | 86.22 |
| *Annotated NR contigs* | 20,338 | 28,499 |

## Gene expression estimation in the liver and gills of the silverside *B. microlepidotus*

The livers of 21 individuals (3–6 per site) and gills of 23 individuals (5–6 per site) were subjected to RNA-seq after an RNA integrity test, resulting in 713,922,789 total raw reads, of which 368,882,159 and 328,822,670 were retained for the gill and liver, respectively, after filtering and trimming. After clustering, 63,424 and 105,131 non-redundant contigs >200 bp were obtained for the liver and gills, respectively. Functional annotation of these non-redundant contigs resulted in 20,338 and 28,499 annotated contigs for the liver and gills, respectively (Table 2).

## Differential expression

RNASeqPower was used to assess the statistical power of this experimental design. The power was estimated to be 0.99 for the liver samples, with an average coverage

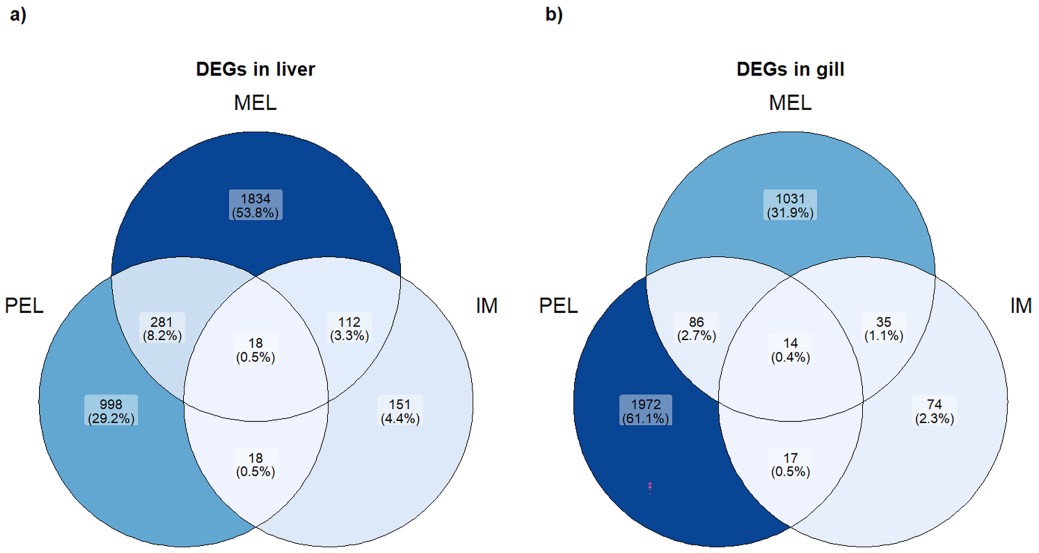

**Figure 3 Venn plot of DEGs in each population and each organ studied.** Number and percentage of DEGs in each population in liver (A) and gill (B).

of 59× and a variation coefficient of 0.20, and 0.90 for the gill samples, with an average coverage of 30× and a variation coefficient of 0.32.

Based on the PCA analysis, three comparisons were performed to identify differentially expressed genes (DEGs): MEL *vs.* SFM, PEL *vs.* SFM, and IM *vs.* SFM. The MEL *vs.* SFM comparison identified 2,245 and 1,166 DEGs for the liver and gill, respectively. The PEL *vs.* SFM comparison identified 1,315 and 2,089 DEGs for the liver and gill, respectively. Finally, the IM *vs.* SFM comparison identified 299 and 140 DEGs for the liver and gill, respectively. The comparisons with the highest DEGs numbers were MEL *vs.* SFM (2,245 DEGs) and PEL *vs.* SFM (2,089 DEGs; Fig. 3). A total of 48 and 37 contigs were identified as CYP genes in liver and gill, respectively. In the case of IM site, three and one contigs identified as CYP genes were differentially expressed in liver and gill, respectively. In the case of MEL site, ten and five contigs identified as CYP genes were differentially expressed in liver and gill, respectively. In the case of PEL site, four and five contigs identified as CYP genes were differentially expressed in liver and gill, respectively. Most contigs identified as differentially expressed CYP genes showed decreased expression in the three polluted sites compared to the SFM non-polluted site, except in the gills at the PEL polluted site where three of five CYP genes were upregulated (Fig. 4). The IM site had the fewest differentially expressed CYP genes, three in liver and one in gill, which could be related to it being a historical reference site, while the MEL site had the most with ten in liver and five in gill (Table 3).

Among the DEGs, seven CYP genes met the criteria of being present in both assembly sets and having a known function. Two (CYP family 4 subfamily B member 1 (CYP4B1) and CYP1A) were differentially expressed in all three polluted sites. Three were classified as endogenous, and four as exogenous based on their function (Table 4). In this group of seven CYP genes, those classified as endogenous showed differential expression in the liver but not in the gills, except for CYP family 26 subfamily B member 1 (CYP26B1).

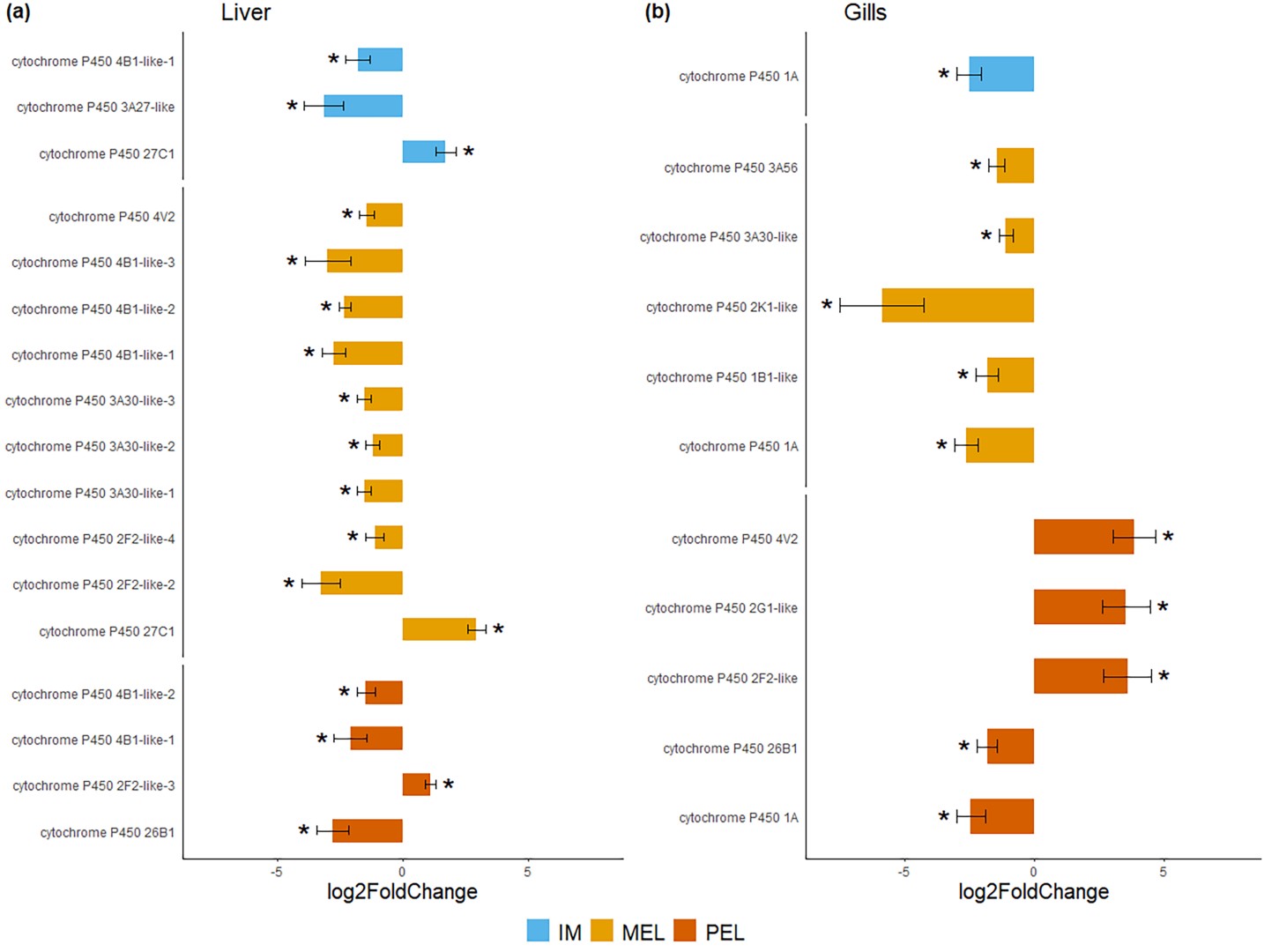

**Figure 4 Bar plot of differentially expressed CYP genes.** Differentially expressed genes (Log₂ fold change > = 1 and FDR < 0.05) in liver (A) and gills (B) expressed as Log₂ fold change compared to gene expression in the SFM population.

In contrast, those classified as exogenous CYP genes showed differential expression in the gills but not in the liver, except for CYP family 2 subfamily F member 2 (CYP2F2). The seven differentially expressed CYP genes, endogenous or exogenous, showed downregulation in all sites and both organs, except for CYP family 27 subfamily C member 1 (CYP27C1; Fig. 5). Additionally, the GLM confirmed the general downregulation of CYP4B1 in the liver (Fig. S1) and CYP1A in the gill (Fig. S2) at all polluted sites. The trend was unclear for the other CYP genes, except CYP2F2 in the liver, which showed a similar pattern to CYP4B1 but was not detected as differentially expressed in IM site.

## DISCUSSION

Our results showed that most CYP genes were not dysregulated, but a small subset was differentially expressed in the studied tissues and within gene family categories.

As expected, the gill and liver exhibited distinct CYP gene expression patterns.

**Table 3 Summary statistics of cytochrome P450 genes with differential expression in liver and gill of *B. microlepidotus*.**

**Liver**

| Comparison | Contig | CYP gene | Log2FoldChange | lfcSE | *p* value | FDR |
|---|---|---|---|---|---|---|
| SFM-IM | comp4372_seq0 | cytochrome P450 27C1 | 1.689140386 | 0.40334787 | 2.8167E−05 | 0.00501397 |
| SFM-IM | comp8176_seq1 | cytochrome P450 3A27-like | −3.161733365 | 0.77290781 | 4.3007E−05 | 0.00656939 |
| SFM-IM | comp904_seq0 | cytochrome P450 4B1-like | −1.802593941 | 0.48467022 | 0.00019984 | 0.01955759 |
| SFM-MEL | comp4372_seq0 | cytochrome P450 27C1 | 2.926656062 | 0.35617173 | 2.0869E−16 | 1.4927E−13 |
| SFM-MEL | comp904_seq0 | cytochrome P450 4B1-like | −2.76255704 | 0.47266937 | 5.0783E−09 | 5.2835E−07 |
| SFM-MEL | comp1968_seq0 | cytochrome P450 3A30-like-1 | −1.544924014 | 0.27106814 | 1.2023E−08 | 1.1052E−06 |
| SFM-MEL | comp1968_seq2 | cytochrome P450 3A30-like-3 | −1.548997777 | 0.27520118 | 1.8168E−08 | 1.5345E−06 |
| SFM-MEL | comp2540_seq0 | cytochrome P450 4V2 | −1.440749922 | 0.2927588 | 8.5977E−07 | 3.6853E−05 |
| SFM-MEL | comp1968_seq1 | cytochrome P450 3A30-like-2 | −1.204941277 | 0.27437655 | 1.1254E−05 | 0.00030868 |
| SFM-MEL | comp312_seq3 | cytochrome P450 2F2-like | −3.275510741 | 0.75196237 | 1.3249E−05 | 0.00035222 |
| SFM-PEL | comp465_seq0 | cytochrome P450 2F2-like | 1.07619613 | 0.22363485 | 1.4921E−06 | 0.00022598 |
| SFM-PEL | comp18298_seq0 | cytochrome P450 26B1 | −2.819234886 | 0.64728693 | 1.3279E−05 | 0.0011892 |
| SFM-PEL | comp904_seq0 | cytochrome P450 4B1-like | −2.103256843 | 0.64399858 | 0.00109104 | 0.02230835 |
| **Gill** | | | | | | |
| SFM-IM | comp1450_seq0 | cytochrome P450 1A | −2.49134759 | 0.4810685 | 2.2334E−07 | 0.00024298 |
| SFM-MEL | comp1450_seq0 | cytochrome P450 1A | −2.59374622 | 0.44172136 | 4.3081E−09 | 1.5424E−06 |
| SFM-MEL | comp2182_seq3 | cytochrome P450 3A56 | −1.43411588 | 0.3216203 | 8.233E−06 | 0.00060507 |
| SFM-MEL | comp9471_seq0 | cytochrome P450 1B1-like | −1.79642649 | 0.43044651 | 3.0008E−05 | 0.00158174 |
| SFM-MEL | comp4170_seq2 | cytochrome P450 3A30-like | −1.07539446 | 0.26904136 | 6.4114E−05 | 0.00275898 |
| SFM-MEL | comp35310_seq1 | cytochrome P450 2K1-like | −5.85419913 | 1.6098108 | 0.00027629 | 0.00819199 |
| SFM-PEL | comp20750_seq0 | cytochrome P450 4V2 | 3.8691694 | 0.80935864 | 1.7483E−06 | 0.0004172 |
| SFM-PEL | comp18949_seq0 | cytochrome P450 26B1 | −1.79835157 | 0.39075461 | 4.1795E−06 | 0.00063625 |
| SFM-PEL | comp1450_seq0 | cytochrome P450 1A | −2.42904691 | 0.55468891 | 1.1916E−05 | 0.00119991 |
| SFM-PEL | comp30438_seq0 | cytochrome P450 2F2-like | 3.61346511 | 0.92393065 | 9.1926E−05 | 0.00500254 |
| SFM-PEL | comp27756_seq0 | cytochrome P450 2G1-like | 3.5519292 | 0.91943001 | 0.00011192 | 0.00564673 |

**Table 4 Selected CYP genes (see Materials and Methods section), type of target compound and function reported in the literature.**

| Gene id | Compound | Function |
|---|---|---|
| CYP26B1 | Endogenous | Inactivation of retinoic acid through oxidation (*Zhao, Dobbs-McAuliffe & Linney, 2005*). |
| CYP27C1 | Endogenous | Catalyzes the transformation of vitamin A1 to vitamin A2 (*Enright et al., 2015*). |
| CYP4B1-like | Endogenous | An endogenous function has been documented, acting over lipids (*Baer & Rettie, 2006*). |
| CYP1A | Exogenous | Metabolization of a wide variety of xenobiotics (*Uno, Ishizuka & Itakura, 2012*). |
| CYP1B1 | Exogenous | Metabolization of resorufin-based compounds and Benzo[a]pyrene (BaP) (*Scornaienchi et al., 2010*). |
| CYP2F2 | Exogenous | Metabolization of naphthalene. Involved in production of potentially toxic intermediate (*Li et al., 2011*). |
| CYP2K1 | Exogenous | Metabolization of lauric acid. Involved in production of carcinogenic form of AFB1(*Yang et al., 2000*). |

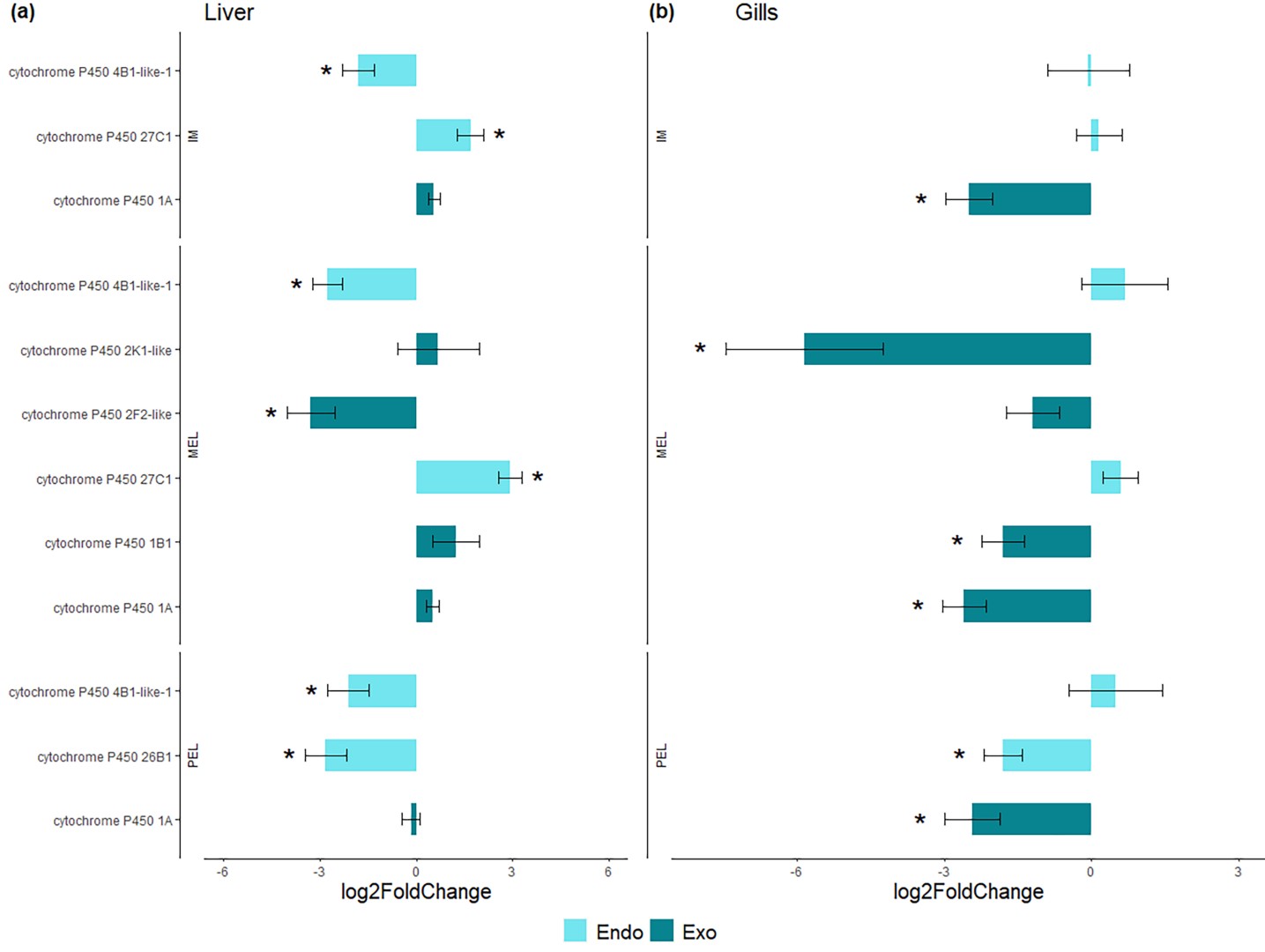

**Figure 5** Bar plot of gene expression of CYP genes that pass the filter in liver (A) and gills (B). Y-axis shows gene names and study site, X-axis shows Log2 fold change compared to SFM population gene expression. CYP genes encoding enzymes acting on exogenous compound are in dark green bars while those acting on endogenous compounds are in light green, in liver (A) and gills (B). CYP genes marked with "*" showed differential expression (Log$_2$ fold change > = 1 and FDR < 0.05). Only CYP genes that meet the criteria of (i) being present in both assemblies and (ii) have known function were considered.

Additionally, the analysis of differential expression conducted on *B. microlepidotus* populations across three polluted sites and one non-polluted site revealed two additional patterns: (i) most CYP genes were not dysregulated, but those differentially expressed were mainly downregulated at polluted sites; and (ii) for the seven CYP genes present in both, liver and gill, those associated with exogenous compounds were differentially expressed in gills, while those associated with endogenous compounds were differentially expressed in the liver.

Organ-related differential gene expression has been well documented. *Yuan et al. (2013)* characterized and measured the expression of five CYP genes in different tissues of the rare minnow (*Gobiocypris rarus*) after BAP exposure. CYP1A, CYP1B, and CYP1C were

strongly upregulated in the liver, gills, and intestine, while CYP2Y3 was only upregulated in the liver (*Yuan et al., 2013*). Populations of the mussel *Mytilus galloprovincialis* exposed to different pollutant levels at six coastal sites around Portugal showed different glutathione S-transferase (GST) gene expression patterns in the gills and digestive gland. This gene is well-known to be related to oxidative stress and is more highly expressed in the digestive gland than in the gills (*Hoarau et al., 2006*). Therefore, the organ chosen for biomonitoring with CYP genes is relevant to interpreting the gene expression pattern accurately.

CYP gene downregulation has been previously reported in the brain of the *Fundulus heteroclitus* fish naturally exposed to persistent toxic chemicals (*Fisher & Oleksiak, 2007*). That study found that two different CYP genes (CYP1B1 and CYP family 2 subfamily N member 2 (CYP2N2)) associated with exogenous compounds were downregulated at all polluted sites, suggesting a possible convergent adaptation to chronic pollution related to reduced procarcinogenic compound activation. In our study, two CYP genes (CYP1A and CYP4B1) were differentially expressed (downregulated) at all polluted sites, suggesting that an adaptive mechanism to chronic pollution (due to wastewater discharge and agriculture activities) could be ongoing in this species at these sites. Another study by *Leaver et al. (2010)* found a similar pattern in the European flounder (*Platichthys flesus*) chronically exposed to coastal sediments with multiple contaminants. They described that hepatocytes from exposed fish had decreased CYP1A expression, a gene typically upregulated by short-term chemical exposure. They attributed this evidence to low pollutant bioaccumulation or poor responsive behavior associated with long-term exposure to polycyclic aromatic hydrocarbons (PAHs; *Leaver et al., 2010*). The decreased CYP gene expression could be associated with global desensitization of the aryl hydrocarbon receptor (AHR) signaling pathway, which is activated by organic xenobiotics such as PAHs and is known to promote CYP1A gene expression in fish (*Zhou et al., 2010*).

This desensitization response has been shown in natural *Fundulus grandis* populations inhabiting polluted environments (*Oziolor et al., 2019*). AHR gene knockdown in zebrafish protected embryos against PAHs (*Billiard et al., 2006*). Studies in killifish (*Fundulus heteroclitus*) exposed to chronic pollution have also shown this AHR desensitization pattern (*Whitehead et al., 2017*). In our study, we find downregulation of CYP1A gene, an AHR regulated gene, in the gills of *Basilichthys microlepidotus* at all polluted sites, suggesting this could be an adaptation to chronic pollution to reduce toxic effects and chemical stress.

The second pattern found in this study was the relationship in response to pollution between tissue and the compound type on which the CYP gene family acts. To our knowledge, this pattern has not been reported before, and we hypothesize that it is related to two fundamental aspects: organ function and adaptation to polluted environments. For our study, we selected two of the most studied organs, the liver and gills, to test pollution's effect on fish, which have shown different responses to chemical pollution, including biochemical, histological, and differential gene expression (*Abdel-Moneim, Al-Kahtani & Elmenshawy, 2012*; *Oliva et al., 2014*). The liver is associated with different metabolic pathways associated with endogenous and exogenous compounds. It has been

shown that CYP26B1 and CYP27C1 are associated with vitamin A metabolism (*Zhao, Dobbs-McAuliffe & Linney, 2005*; *Enright et al., 2015*). Vitamin A is an important micronutrient stored as retinol in the liver, participating in different biological processes such as vision and cell differentiation and proliferation throughout the life of organisms (*Hernandez & Hardy, 2020*). It is well documented that organic pollutants affect the retinol status in the liver of exposed fishes, where a high concentration of these pollutants depletes the retinol content (*Rolland, 2000*).

Conversely, fatty acid metabolism is also affected by pollution (*Yousafzai & Shakoori, 2011*), potentially related to CYP4B1 activity, whose gene was downregulated in our study. The gills are an important organ since they are the closest contact between a fish and its environment. Here, we found that from the seven CYP genes that show differential expression those related to exogenous compounds were downregulated in the gills, potentially related to AHR pathway desensitization, as mentioned above for the liver. The gills play an important role in exchanging some molecules with the environment and facilitating the interaction with pollutants, affecting both gill morphology and physiology (*Evans, Piermarini & Choe, 2005*). After seven days of exposure to Cd, *Danio rerio* gills showed upregulation of genes associated with the oxidative stress response and mitochondrial metabolism. However, the expression of most of these genes decreased to their basal level after 21 days of exposure (*Gonzalez et al., 2006*). Similarly, *Mustafayev & Mekhtiev (2014)* found that CYP genes were downregulated in the gills of *Alburnoides bipunctatus* inhabiting polluted rivers. In addition, *Whitehead et al. (2012)* found CYP1A upregulation in the gills of *Fundulus heteroclitus* exposed to the Deep-Water Horizon event, a major oil spill.

An important point in this second pattern is the classification of CYP genes related to their function (endogenous *vs*. exogenous compounds). According to *Uno, Ishizuka & Itakura (2012)*, some gene families act on exogenous compounds (*e.g.*, CYP1), and others act on endogenous compounds (*e.g.*, CYP27) in fish. This pattern is also argued by *Burkina et al. (2021)*. However, recent studies show that some families that act on exogenous compounds also have endogenous targets (*Larigot et al., 2022*) and *vice versa* (*Röder et al., 2023*) in other vertebrates. Therefore, it is important to consider more studies in fish to confirm the target compounds of the different CYP families.

Our results showed that CYP1A was downregulated at polluted sites. We hypothesize that this results from chronic exposure to pollutants rather than an acute effect, as described for species exposed to pollutants for short durations (*Yuan et al., 2013*). The gene expression pattern observed in our study is similar to that observed in natural *Fundulus heteroclitus* populations exposed to chronic pollution (*Fisher & Oleksiak, 2007*). However, this pattern could be related to the pattern observed for *Danio rerio* after several days of exposure described above (*Gonzalez et al., 2006*), leading to basal gene expression, which is far from the upregulation observed in acute exposure (*Yuan et al., 2013*). Our results suggest an association between tissue sorting and the function of CYP family genes in response to pollution. This association reveals a distinction between CYP genes related to endogenous and exogenous compounds. Therefore, the observed expression pattern could impact the interpretation of CYP gene expression as a biomarker of pollution.

The CYP genes have received significant attention as pollution biomarkers, particularly CYP1A. However, this discussion has been around elements such as expression threshold or biomarker classification, in most cases associated with upregulation, regardless of the chronic responses in this gene family (*Oris & Roberts, 2013*). Overall, our results in chronically polluted natural populations suggest that environmental impact studies should focus on the organ and CYP gene studied and their interactions.

## CONCLUSIONS

Most of the CYP genes detected in this study did not present differential expression. However, in the seven CYP genes that did show variation, downregulation was detected in the polluted sites, three of them are differentially expressed in the gills and act on exogenous compounds, while the four differentially expressed in the liver act on endogenous compounds.

The downregulation detected suggested adaptation to chronic pollution environments, as has been suggested before for other species with similar CYP response pattern, while the differential expression of genes acting on endogenous or exogenous compounds could be related to the organ function, with gill being a more exposed organ interacting closely with exogenous compounds and liver as an organ responsible of many metabolic pathways and related to many endogenous compounds. Overall, our study suggests the existence of an interaction between gene family and tissue in the gene expression response to pollution, then, it is necessary to take this into account for biomonitoring in chronically polluted environments with CYP genes.

## ACKNOWLEDGEMENTS

Thanks to R Gauci and Matias Briones for help during field work.

### Funding

This work was supported by the Fondo Nacional de Desarrollo Científico y Tecnológico (11150213). David Veliz was supported by the Chilean Millennium Initiative grant ESMOI. Jorge Cortés-Miranda was supported by Agencia Nacional de Investigación for Doctoral Fellowship (21200769) and doctoral thesis fellowship (242220080). The APC was funded by the UTAMayor 9737-23 and Nucleo Milenio INVASAL NCN2021-056. The funders had no role in study design, data collection and analysis, decision to publish, or preparation of the manuscript.

### Grant Disclosures

The following grant information was disclosed by the authors:
Fondo Nacional de Desarrollo Científico y Tecnológico: 11150213.
Chilean Millennium Initiative grant ESMOI.
Agencia Nacional de Investigación: 21200769 and 242220080.

UTAMayor: 9737-23.
Nucleo Milenio: INVASAL NCN2021-056.

## Competing Interests
The authors declare that they have no competing interests.

## Author Contributions
- Jorge Cortés-Miranda conceived and designed the experiments, performed the experiments, analyzed the data, prepared figures and/or tables, authored or reviewed drafts of the article, and approved the final draft.
- Noemí Rojas-Hernández performed the experiments, authored or reviewed drafts of the article, and approved the final draft.
- Gigliola Muñoz performed the experiments, analyzed the data, authored or reviewed drafts of the article, and approved the final draft.
- Sylvia Copaja conceived and designed the experiments, authored or reviewed drafts of the article, and approved the final draft.
- Claudio Quezada-Romegialli conceived and designed the experiments, authored or reviewed drafts of the article, and approved the final draft.
- David Veliz conceived and designed the experiments, prepared figures and/or tables, authored or reviewed drafts of the article, and approved the final draft.
- Caren Vega-Retter conceived and designed the experiments, prepared figures and/or tables, authored or reviewed drafts of the article, and approved the final draft.

## Animal Ethics
The following information was supplied relating to ethical approvals (*i.e.*, approving body and any reference numbers):

Ethics committee of the Universidad de Chile and complied with existing laws in Chile (Resolución Exenta No. 3078 Subsecretaria de Pesca).

## Field Study Permissions
The following information was supplied relating to field study approvals (*i.e.*, approving body and any reference numbers):

Sampling at Maipo River was approved by Subsecretaría de Pesca y Acuicultura de Chile (Resolución Exenta No. 3078 Subsecretaria de Pesca).

## Data Availability
The data for RSEM counts of differentially expressed CYP genes in liver and gill tissue, and physical and chemical data are available at the repository of Universidad de Chile: Cortés-Miranda, Jorge; Rojas-Hernández, Noemi; Muñoz, Gigliola; Copaja, Sylvia; Quezada-Romegialli, Claudio; Veliz, David; Vega-Retter, Caren, 2022, "La selección de biomarcadores depende de la función genética y del órgano: el caso de los genes de la familia del citocromo P450 en peces de agua dulce expuestos a contaminación crónica",

The raw sequencing data is available in GenBank and BioSample: PRJNA1033453: SAMN38033975–SAMN38034018.

## Supplemental Information

Supplemental information for this article can be found online at http://dx.doi.org/10.7717/peerj.16925#supplemental-information.

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
