# Peer review of "Biomarker selection depends on gene function and organ: the case of the cytochrome P450 family genes in freshwater fish exposed to chronic pollution"

_PeerJ, doi:10.7717/peerj.16925_

## Round 0.1 · original submission · Major Revisions

The manuscript received three thorough reviews from experts in the field. Two of the reviewers recommended major revisions and one recommended that I reject the paper. I believe that the paper could potentially become acceptable for publication in the journal, so I have recommended Major Revisions.

Please pay attention to all of the reviewer comments and try to address them or explain why you are not addressing them.

From my perspective, some of the key concerns raised by the reviewers include:

- the focus on a small subset of CYP genes, without showing all differentially expressed CYP genes

- the classification of CYP genes as acting on “endogenous” or “exogenous” substrates is not as clear-cut as presented in the paper. The authors should provide much more information on how this classification was made and acknowledge the challenges in doing such a classification

- the lack of information about other altered expression of other genes and pathways

- the need to deposit the data in a publicly accessible database

- the lack of information about organic contaminants, which is the primary class of contaminants causing altered expression of many CYPs

- a more rigorous statistical analysis and presentation of the data

- more information about why IM was at first considered a non-polluted site and then a polluted site.

Reviewer 1 ·

Basic reporting

This study aimed to elucidate CYP gene expression patterns in different tissues using the silverside Basilichthys microlepidotusas as a model under chronic pollution exposure. The results showed that the expression of CYP gene in various tissues was different, and genes and tissues were important factors to consider when using biomarkers to evaluate chronic pollution. The authors did a lot of work in determining many sorts of parameters, but not comprehensively showed or explained the major relevant ones at least in just numeric form. I recommend the manuscript needs significant revision before it can be accepted, several aspects as follows should be revised or clarified.

General:
1. The introduction seems longer than most articles, please try to make it concise.
2. As you have determined the concentration of metals, etc., of environmental matrices, please list the concentration of metals and other major contaminants to add the significance of this manuscript.
3. Physical and chemical characterization should be split to separate subsections, e.g. the water/sediment, or by different classes of distinct parameters.
4. Add the data and discussion of the pollution condition of each site. Were they greater than the standards?
5. Add additional data and discussion about the whole transcriptome, especially about the pollution-induced pathway alteration.

Specific:
1. Line 112 Why did you choose Basilichthys microlepidotusas? Did you compare it with other kinds of fish, was it highly contaminated?
2. Line 128 Are the sampling sites and number of samples insufficient, and will the size, age, and sex of the fish have an impact on the results? What is the environmental pollution condition of each site, please give some basic information about that.
3. Line 130 Give the length and weight of the fish of each group.
4. Line 300 Please list all the differentially expressed CYP genes of at least one pair of comparisons in a table in the manuscript, and add others in the SI.
5. Line 130 Why choose the gill and the liver regardless of other tissues, such as the intestine, the kidney, and the brain?
6. Line 224 was the sequencing data deposited in any well-known database, please list them in the manuscript.
7. Line 298 please give a Venn plot to show the distribution of DEGs of each site.

Experimental design

no comment

Validity of the findings

no comment

Reviewer 2 ·

Basic reporting

The MS entitled “Biomarker selection depends on gene function and organ: the case of the Cytochrome P450 family genes in freshwater fish exposed to chronic pollution” describes partial results from an RNA-seq-centered investigation using fish liver and gill sampled from a reference and three pollutant sites in the same river. From the thousands of transcripts sequenced and annotated, authors focused their analysis on the seven CYP genes differently expressed between at least one of the polluted sites and the reference site and identified “three different patterns: (i) most CYP genes were downregulated in polluted sites; (ii) the gill and liver had different CYP gene expression patterns; and (iii) CYP gene families associated with exogenous compounds were differently expressed in gills, while those associated with endogenous compounds were differently expressed in the liver” (lines 321 – 324).

The first pattern (i - most CYP genes were downregulated in polluted sites) is incorrect. Most CYP genes were not regulated, as they were not identified as differently expressed. The second identified pattern (ii - the gill and liver had different CYP gene expression patterns) is far from original. It is widely known that different organs have different gene expression profiles; that is among the reasons why they are different organs. The third pattern (iii – CYP genes associated with endogenous Vs exogenous compounds) is highly controversial as a single CYP gene frequently encodes proteins displaying different catalytic functions, acting on different substrates that could be of endogenous or exogenous origin. For example, Annu. Rev. Pharmacol. Toxicol. 2022. 62:383–404 and THE JOURNAL OF BIOLOGICAL CHEMISTRY VOL.283, NO.52, pp.36061–36065) shows endogenous functions/substrates for CYP1A, while Int. J. Mol. Sci. 2023, 24 shows CYP4B1 on a diversity of xenobiotics furans and cyclic amines.

At some point of the MS (ln 331 – 332), authors considered “a convergent adaptation to chronic pollution” a possible explanation for the observed downregulation of CYP1A and CYP4B1, similarly to what has been widely described for some Fundulus heteroclitus populations. However, while those Fundulus populations lives in habitats chronically contaminated with industrial organic contaminants (e.g. PCBs and HPAs), the fish used in this work inhabits a basin contaminated mostly by “domestic and agricultural activities” (ln 93 – 94).

The research behind the MS is of great value, but the author’s decision to focus their analysis solely on the seven regulated CYP genes, making this controversial separation between associated with endogenous or exogenous compounds, and their expression pattern in gill and liver led to flawed conclusions.

I strongly suggest a re-analysis of the complete RNA-Seq dataset and the elaboration of a new MS.

Minor issues:

1 – review English. For example: ln 116 – “Due to his [should be its] use as a biomarker …”;
2 – sex of the individuals: that might become an important issue. Six is a reasonable amostral, number but if all samples are replicates and as you cannot guarantee they are all from the same sex…trouble…specially in the groups you lost few samples to due low RIN number;
3 – No organic industrial contaminants were measured and those are in general the most potent regulators of CYP genes involved in the metabolism of xenobiotics;
4 – Ln 347 unpublish data -> I believe the work would be stronger if all available RNA-Seq data were analyzed;
5 – the bulk reads should be deposited at a public database, for example NCBI SRA;
6 – Figure 2: describe in the legend what is A and what is B. It is specified in the text (ln 269) but should be in the figure legend too;
7 – Is there truly a reason to show both Figure 3 and Figure 4? In fact, they appear to have some conflict. For example, Figure 3 shows 7 differently expressed CYPs in the liver of MEL fish but Figure 4 shows only 3 differently expressed CYPs in the liver of MEL fish. This conflict is true for all groups, except gills of IM fish that has only 1 differently expressed CYPs in both figures;
8 – The number of clean reads from liver differs in table 1 from what is written in the text ln 284.

Experimental design

'no comment'

Validity of the findings

'no comment'

Additional comments

'no comment'

Reviewer 3 ·

Basic reporting

The comparison of cyp expression differences across different polluted locations in relation to a known reference site and influence of tissue type provides an interesting perspective for this species, the neotropical silverside. Although the premise of this study is interesting, it wasn’t clear how these transcript profiles were compared across and between sites. The expression relationships are not presented in a way to fully understand if there are significant differences in the same cyp across sampled populations, or if a site just had a handful of genes that were differentially expressed. To best answer this question, the DEGs from each CYP targeted should further be compared, relative to the reference location, using an ANOVA. As presented now, there really is no indication that there are strong/any differences between the contaminated sites, aside from the reference itself.

Experimental design

Line 53- Phenotypes aren’t identified with omics techniques, just the genes/metabolites that may influence the response. This should be clarified or removed.
Line 61- This wording is a bit awkward and suggest changing to something like, “The expression of genes have been…”. Using differential and differ in the same sentence isn’t recommended, also because genes are likely to be expressed differently, but I feel the real statement to be made is either that there are significantly differentially expressed genes are that these genes are being dysregulated.
Line 105- Considering these acronyms are different from the ones used on lines 123-127, I suggest not including them here, as it is a bit confusing and not used elsewhere. I recommend just keeping the acronyms from lines 126-128.
Line 110- Change to “differentially expressed”.
Line 111- Polluted sites relative to the non-polluted reference sites?
Line 116- Change “his” to “the” possibly?
Line 128- Is there an indication of size/age that could be provided for context?
Line 139- How were water samples collected? What were they stored in? At what temperature were they stored at and for what duration?
Line 140- Electrical conductivity should be defined before acronym used, instead of line 258.
Line 177- A total of 10 mL of solution what initial volume of water? Was this normalized by weight or volume?
Line 223- Which GO groups were considered (BP, CC, MF)?
Line 231- What version of RSEM?
Line 246- Function known in general or in fish, specifically?

Validity of the findings

Lines 253-254- Considering that IM was eventually considered as a contaminated site for this study, it is important to note that “Historically, MEL and PEL were considered to be polluted sites and SFM and IM as non-polluted”, or something of the like.
Lines 258-262- As the acronyms for all of these targeted parameters were used before, the full name should be used previously in the methods section, lines 141-142, and at first use for ones not provided on these lines.
Line 271- Change “has” to “had”.
Line 273- Lowercase The, since starting with the 2016 data.
Lines 301-303- Were these significantly differentially expressed genes? If so, an asterisks should be placed within Fig 3.
Line 303- Change “were” to “where”.
Lines 303-305- Potentially because it used to be considered a reference site?
Lines 313-314- Were these significantly dysregulated, or trends? If not significant, is there a “true” differing response between the populations?

---

## Round 0.2 · Minor Revisions

Thank you for your revisions, which have clarified some but not all of the reviewers' concerns. Please pay close attention to the new reviewers' comments.

Rev 1 raises several important points that should be easily addressed.

Rev 2 raises some of the same concerns as in their initial review. You have clarified some of the issues they raised, but there are still some parts of the results and discussion that seem misleading. I suggest paying special attention to the points (ii) and (iii) raised by this reviewer as well as the phrasing in lines 34 ("Most CYP genes..."), 326 ("Most endogenous CYP genes...", 396 ("...most CYP genes related to exogenous compounds...", and any other places where the wording lacks the qualifications related to the small subset of genes you looked at.

In addition to the reviewers' comments, I have a few questions that you might clarify in your revision.
1) In Fig 4, there are three different CYP4B1-like genes listed, whereas in Fig 5 only one CYP4B1-like is listed. What is the relationship between the three CYP4B1-like genes listed in Fig 4 to the single gene listed in Fig 5?
2) Line 366: What is meant by “AHR-related genes”? The CYP genes examined in this paper are not “AHR-related”, although some of them are known to be regulated by AHR. Or are you referring to another study? If so, please cite it.

Reviewer 1 ·

Basic reporting

The author has revised the manuscript according to the suggestions, however, there are still minor revisions that should be done before it can be accepted, as listed below:

Line 37:This sentence is hard to understand.

Line 150: Could the non-metal ions(e.g., the nitrite, carbonate, and sulfate) be detected directly by atomic absorption spectroscopy? As we do not see the methods reported by Clesceri et al., please clarify this.

Line 157: Remove the '-' behind the HNO3.


Line 164: Remove the '2-' behind the MnSO4. Please check the entire manuscript.


Line 200: Some should be ions e.g., Na, K, etc.


Line 205: Please declare the accessions or the link for where you deposited the raw sequencing data.


Table 1: use decimal point instead of comma.

Experimental design

no comment

Validity of the findings

no comment

Additional comments

no comment

Reviewer 2 ·

Basic reporting

The revised version of the MS entitled “Biomarker selection depends on gene function and organ: the case of the Cytochrome P450 family genes in freshwater fish exposed to chronic pollution” does not address to well the main three issues that were raised.

(i) most CYP genes were downregulated in polluted sites: indeed authors rephrased the sentences to specify that they refer specifically to the CYP genes that were differently expressed, but the text still passes the idea that the downregulation was the main effect while the main effect was non-regulation. Only quite few CYP genes were actually regulated. As far as I could find the actual number of regulated CYP genes was only given to an exception case (line 320) when only five CYP genes were regulated, three of those up-regulated. I do believe the writing could misguide the readers to get a false impression.

(ii) the gill and liver had different CYP gene expression patterns; the comment made in the first version remain. This is absolutely expected. In this version authors included some discussion about the regulated CYP genes being related to endogenous or exogenous compounds, but that only turned the conclusion more controversial (as it will be addressed later). Indeed, the question raised in (i) is also true here. When authors say "most" they are referring to most of a total of seven CYP genes that met their criteria (ln 323). From the seven regulated CYP that met author's criteria three were classified as endogenous and four as exogenous (ln 325 - 326). So again most of the CYP genes were not regulated and were not classified as endogenous or exogenous.

(iii) CYP gene families associated with exogenous compounds were differently expressed in gills, while those associated with endogenous compounds were differently expressed in the liver” (lines 321 – 324): Again From the seven regulated CYP that met author's criteria three were classified as endogenous and four as exogenous (ln 325 - 326). So again most of the CYP genes were not regulated and were not classified as endogenous or exogenous. I believe the conclusion is based upon weak evidence and that the writing is too bold and could misguide readers. Moreover, the cited reference of DOI 10.1016/j.etap.2012.02.004 does not make such a clear cut between CYP with endogenous and exogenous functions. For example the work endogenous is only found four times in the paper, while the word exogenous is only found three times. The abstract does not refer to any classification of CYP families into endogenous or exogenous and the paper only "pinpointed" (using the word used in the original cited paper) 8 of the 18 CYP families.

The comment made in the first round of revision remain valid: the work is relevant and well performed, the authors decision to slice the results and publish only the results regarding CYP genes is controversial but acceptable, but the comparisons / contrasts (endogenous X exogenous | liver X gill) made are troublesome. This became clear when analyzing the conclusions:

"Our study suggests a different association of CYP gene expression and tissue in the silverside B. microlepidotus." --> This first sentence of the Conclusion does not really say anything. What is exactly "different association"?

"While CYP genes differentially expressed in the gills act on exogenous compounds, those differentially expressed in the liver act on endogenous compounds." --> Three act on exogenous while four act on endogenous, or opposite. The point here is that they are quite small numbers. Most of the CYP genes were not regulated. Another point here is that this division is quite controversial. Even the referred paper does not make such a clear cut. Once more most of the differently expressed CYP genes were not classified into endogenous or exogenous.

"The differentially expressed CYP genes generally showed downregulation." --> Three act on exogenous while four act on endogenous, or opposite. Most were not regulated.

"These patterns seem related to organ function and adaptation to chronic pollution environments, respectively." --> How?

"Overall, our study emphasizes that caution is needed when using a CYP gene for biomonitoring in chronically polluted environments, where the gene, tissue, and their interaction are relevant." --> Sure thing.

Experimental design

great

Validity of the findings

great

Reviewer 3 ·

Basic reporting

Addressed well.

Experimental design

Addressed well.

Validity of the findings

Addressed well.

Additional comments

All of my previous concerns/suggestions have been incorporated well in the revised version of this manuscript. I do not have further feedback to the authors to improve the manuscript.

---

## Round 0.3 · accepted · Accept

Thank you for your revisions. The manuscript is now acceptable for publication. One additional change that you should make in the production stage is to change the colors mentioned in the legend to Figure 5 (dark green and light green) to match the colors actually used in the figure (red and blue).